# PARAMETER-EFFICIENT DETOXIFICATION WITH CONTRASTIVE DECODING

## ABSTRACT

The field of natural language generation has witnessed significant advancements in recent years, including the development of controllable text generation techniques. However, controlling the attributes of the generated text remains a challenge, especially when aiming to avoid undesirable behavior such as toxicity. In this work, we introduce *Detoxification Generator* (DETOXIGEN), an inference-time algorithm that steers the generation away from unwanted styles. DETOXIGEN is an ensemble of a pre-trained language model (*generator*) and a *detoxifier*. The *detoxifier* is trained intentionally on the toxic data representative of the undesirable attribute, encouraging it to generate text in that style exclusively. During the actual generation, we use the trained *detoxifier* to produce undesirable tokens for the *generator* to contrast against at each decoding step. This approach directly informs the *generator* to avoid generating tokens that the *detoxifier* considers highly likely. We evaluate DETOXIGEN on the commonly used REALTOXICITYPROMPTS benchmark (Gehman et al., 2020) with various language models as *generator*s. We find that it significantly outperforms previous approaches in detoxification metrics while not compromising on the generation quality. Moreover, the *detoxifier* is obtained by soft prompt-tuning using the same backbone language model as the *generator*. Hence, DETOXIGEN requires only a tiny amount of extra weights from the virtual tokens of the *detoxifier* to be loaded into GPU memory while decoding, making it a promising lightweight, practical, and parameter-efficient detoxification strategy.

## 1 INTRODUCTION

Large language models (LLMs) have demonstrated remarkable promise in various generative tasks by first self-supervised pretraining on large text corpora and then finetuning with instruction data for alignment (Mishra et al., 2022). Yet a wealth of previous work has demonstrated that pre-trained models inherit toxicity and biases from their training corpora (Zhao et al., 2019; May et al., 2019; Kurita et al., 2019; Basta et al., 2019). As a result, generative models (OpenAI, 2023; Touvron et al., 2023; Nijkamp et al., 2023) tend to degenerate into unsafe text even when conditioning on seemingly innocuous prompts (Wallace et al., 2019; Sheng et al., 2019; Gehman et al., 2020), which is difficult to resolve by prompt engineering alone (Zong & Krishnamachari, 2022; Liu et al., 2022b; Webson & Pavlick, 2022; Lou et al., 2023).

To address this challenge, a plethora of approaches have been proposed, which usually require full-model finetuning of the underlying language model to build the *detoxifier* (Dathathri et al., 2019; Gururangan et al., 2020; Krause et al., 2021; Liu et al., 2021a). However, nowadays the largest LLMs typically contain more than 100 billion parameters, making such resource-intensive tuning less viable. This trend calls for more parameter-efficient approaches.

In this work, we propose DETOXIGEN (Figure 1), a parameter-efficient framework that leverages the frozen weights of the language model itself and only introduces a tiny portion of new model parameters to detoxify generation.[1] During training, we use prompt tuning (Lester et al., 2021) to train a *detoxifier* exclusively on toxic data with the next-token prediction objective. The resulting *detoxifier* shares all the backbone model weights with the LLM (i.e., the *generator*). During inference,

---

[1]We will make our code publicly available.

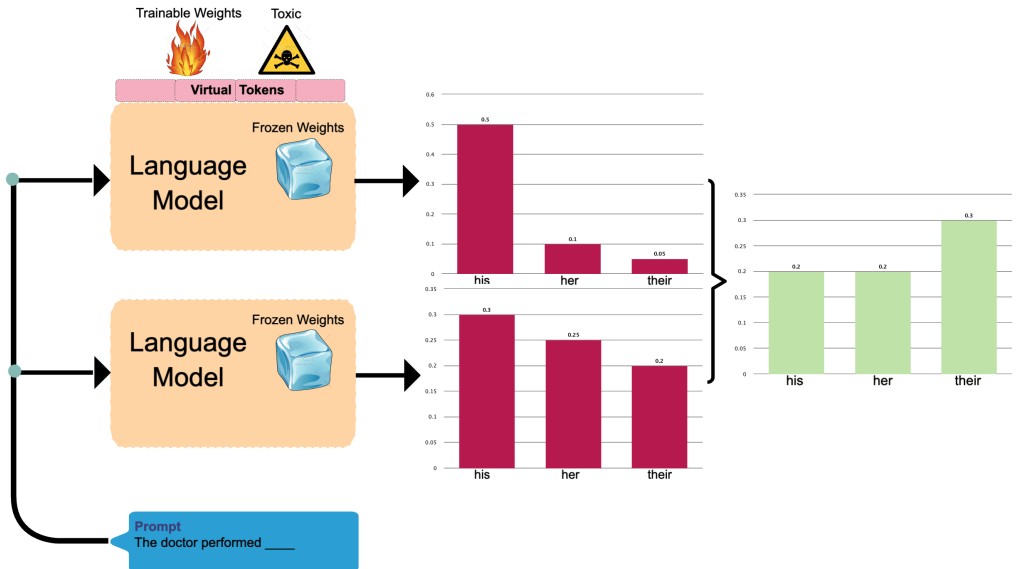

Figure 1: Illustration of the DETOXIGEN pipeline that avoids generating a gender-biased next token. A prompt is fed into both the *generator* and the *detoxifier*, which share the same underlying frozen weights from the backbone language model. Additionally, the *detoxifier* contains virtual tokens whose embeddings are trainable. Such virtual tokens steer the *detoxifier* toward generating only toxic continuations. Each model provides its own probability distribution for the next token, where DETOXIGEN combines the two distributions and performs the detoxification.

we build on top of the contrastive decoding (Li et al., 2023) paradigm and employ the *detoxifier* to manipulate the output probability distribution of the LLM for each generation step. Intuitively, the *generator* avoids outputting tokens that the *detoxifier* considers highly probable. For example, in figure 1 the *detoxifier* considers the gender-biased word "his" very likely as the next token, helping the *generator* to score down the probability of that token.

We evaluate our framework on the REALTOXICITYPROMPTS dataset (Gehman et al., 2020) and find that it outperforms previous approaches on the standard benchmark metrics by a significant margin, indicating that the text generated by our model is both safer and of higher quality. We also conduct ablation studies and pair models of different sizes from the same model family (e.g., the Llama-2 (Touvron et al., 2023) family). These studies show that pairing a *generator* with a *detoxifier* that shares the same backbone LLM is indeed the best-performing configuration.

Our main contributions are: (1) Performance: Propose a detoxification framework that outperforms previous models by a large margin on commonly used detoxification benchmarks/metrics; (2) Efficiency: We apply parameter-efficient learning to controllable text generation for detoxification. Our model introduces the least amount of additional parameters (hence also requires less data to train) as compared to state-of-the-art models; (3) Transferability: Our *detoxifier* model only requires toxic data and does not require any contrastive (non-toxic) data, making our approach transferable thanks to the easier and more manageable data curation.

## 2  MODEL

### 2.1  TASK FORMULATION

We consider controlled decoding-time approaches for open-ended text generation. A *generator*, in our case a language model, receives an unfinished input text as a prompt and aims to output a fluent and coherent continuation that avoids toxicity with the help of a *detoxifier*, which is another language model trained with data of the target attribute.

## 2.2 Model Components

**Generator** Let $x_{<t} = x_1, x_2, ..., x_{t-1}$ be a prompt consisting of $(t-1)$ tokens, where each $t_i(1 \leq i \leq t-1)$ is a token in the vocabulary set $V$ of the language model (LM). The LM encodes $x_{<t}$ in an autoregressive fashion and outputs $\mathbf{z}_t \in \mathbb{R}^{|V|}$, where $\mathbf{z}_t$ denotes the logits for the $t$th token $x_t$ and $|V|$ corresponds to the vocabulary size. The LM then obtains a probability distribution $P_{GEN}$ over $V$ by computing the softmax of $\mathbf{z}_t$

$$P_{GEN}(x_t|x_{<t}) = \text{softmax}(\mathbf{z}_t), \tag{1}$$

and the next token is sampled from this distribution.

**Detoxifier** The *detoxifier* takes as input the same prompt fed to the *generator* for each generation step and computes a probability distribution $P_{CON}(X_t|x_{<t})$ over $|V|$ in the same way. However, the *detoxifier* is not a vanilla LM like the *generator*, but rather an LM specially trained to output toxic content. Intuitively, the *generator* is discouraged from outputting tokens that the *detoxifier* considers highly likely, thus avoiding toxic generations. In other words, the decoding process involves an ensemble of the two LMs to obtain the final output probability distribution $P(x_t|x_{<t})$:

$$P(x_t|x_{<t}) = P_{GEN} + \alpha \Delta P \tag{2}$$
$$\Delta P = P_{GEN} - P_{CON}, \tag{3}$$

where the hyperparameter $\alpha$ denotes the *control strength* of the model and $\Delta P$ represents the *probability correction term* determined by the difference between the two distributions. Intuitively, $\alpha$ dictates how much we want to modify the *generator*'s probability distribution through the correction term $\Delta P$. Since it is possible that $P(x_t|x_{<t})$ contains values that fall out of the $[0.0, 1.0]$ range, making them invalid probabilities, we also clip them on both sides – i.e., setting any value that is below $0.0$ to $0.0$ and any value above $1.0$ to $1.0$. The resulting probability vector is then normalized by first computing its log probabilities and then taking the softmax. Our formulation is closely related to that in Contrastive Decoding (Li et al., 2023) and DExperts (Liu et al., 2021a), but we differ by manipulating in the probability space rather than the logits space because we found in our initial experiments that directly dealing with probability distributions result in better performance on the downstream task.

**Sampling** To constrain the model to only generate plausible tokens, we first employ Nucleus (Top-$p$) Sampling (Holtzman et al., 2020) to limit the vocabulary $V$ to a subset $V^{(p)}$ by only selecting the highest probability tokens whose cumulative probability mass exceeds some threshold $p \in [0.0, 1.0]$. More specifically, given the distribution $P_{GEN}(x_t|x_{<t})$ in Equation 1, the top-$p$ vocabulary $V^{(p)} \subseteq V$ is defined by the smallest vocabulary set such that

$$\sum_{x \in V^{(p)}} P_{GEN}(x|x_{<t}) \geq p. \tag{4}$$

The Top-p sampling then truncates the less reliable tail of the distribution by setting

$$P'[x] = \begin{cases} P[x], & \text{if } x \in V^{(p)} \\ 0, & \text{otherwise.} \end{cases} \tag{5}$$

The *detoxifier* then comes in and only manipulates logits in the set $V^{(p)}$ so that regardless of how $P_{GEN}$ is modified, the generated tokens are guaranteed to be plausible as evaluated by the *generator*. When applying this restriction, equation 1 becomes

$$P'(x_t|x_{<t}) = P'_{GEN} + \alpha(P'_{GEN} - P'_{CON}). \tag{6}$$

## 2.3 PARAMETER-EFFICIENT TRAINING OF DETOXIFIER

As discussed in Section 1, due to the increasing size of large language models, we aim to introduce as few additional model parameters as possible to our framework. Hence we adopt Prompt Tuning (Lester et al., 2021), a parameter-efficient training method, to train a language model exclusively on toxic data. This method learns soft prompts (or virtual tokens), whose embeddings are trainable parameters to condition frozen language models to perform the target downstream tasks.

## 3 EXPERIMENTAL SETUP

### 3.1 BACKBONE MODELS

Following Liu et al. (2021a), we use GPT2-large (Radford et al., 2019) as the *generator* and the backbone of the *detoxifier*. We use GPT2-XL to evaluate the generation quality. For ablation studies reported in Section 4.3, we also consider GPT2 with small and medium sizes.

It is worth noting that previous work selected the GPT2 family mostly because it was one of the strongest models at the time. To observe if the same trend of performance holds for the most recent LLMs, we also experiment with another family of Transformer-based (Vaswani et al., 2017) language models, namely Llama-2 (Touvron et al., 2023) because it satisfies the following three criteria: (1) It is publicly released so that it is easier for researchers to reproduce and compare with our work; (2) It achieves state-of-the-art performance on diverse benchmark datasets (Nijkamp et al., 2023); (3) It has three sizes so that we can evaluate whether larger models can be paired with smaller ones for detoxification – such is the setting when we prioritize reducing latency over minimizing GPU memory footprint. Hence we experiment with Llama-2 with 7B, 13B, and 70B parameters, respectively. Due to the large size of Llama-2-70B, for all our experiments we use bfloat16 for both training and inference to increase throughput and reduce GPU memory usage. We evaluate perplexity from the Llama-2 family with Llama-2-7B unless otherwise stated.

### 3.2 TRAINING OF DETOXIFIER

We prompt tune the *detoxifier* with the standard language modeling objective (Bengio et al., 2000) which learns the parameters of the conditional probability distribution of the next word given the preceding context. We extract the training data from the human-annotated Jigsaw Unintended Bias in Toxicity Classification (Borkan et al., 2019). An example is considered toxic if more than $50\%$ of the annotators classify it as toxic. This threshold splits the corpus into around 160K toxic and $1.4M$ nontoxic examples. We only train our models with the toxic part of the data.

For prompt tuning, we use $100$ virtual tokens for each model with a learning rate of $0.1$. To efficiently explore different parameter-efficient methods, we use the PEFT (Parameter-Efficient Fine-Tuning), a library that wraps around HuggingFace Transformers (Wolf et al., 2020) model objects and provides out-of-the-box implementations for widely adopted PEFT approaches (Mangrulkar et al., 2022). Because we need to obtain logits from the *detoxifier* for each generation step, we overwrite the PEFT model object to only prepend virtual tokens to the input for the first generation step.

### 3.3 HYPERPARAMETER TUNING

We tune the hyperparameter $\alpha$ with a held-out validation set and perform a grid search from $1.0$ to $9.0$ with a $1.0$ increment. As will be shown in Section 4.1, we find that $\alpha = 5.0$ strikes the best balance between toxicity and generation quality. We thus adopt this value throughout all experiments.

### 3.4 EVALUATION DATA

We follow Liu et al. (2021a) to use the REALTOXICITYPROMPTS dataset (Gehman et al., 2020) which contains 100K naturally occurring, sentence-level prompts derived from a large corpus of English web text. These prompts are annotated with toxicity scores and language models are known to degenerate into toxic continuation when conditioning on them. To determine the *detoxifier* strength $\alpha$, we randomly sample 1k prompts as the validation set and another disjoint 10k as the test set.

Table 1: Results on a random nontoxic 10K sample from the REALTOXICITYPROMPTS dataset. On the first row, the downward arrows indicate "the lower the better", while the upward ones indicate the opposite. Avg. Max. Toxicity stands for "Average Maximum Toxicity", PPL stands for "Perplexity", and all models are evaluated with GPT2-XL. Dist-N stands for the Distinct-N metric. All models in this table use GPT2-large as the backbone model, except for the last row where Llama-2-7B is used. State-of-the-art results are boldfaced.

| Model | Toxicity ($\downarrow$) | | Fluency ($\downarrow$) | Diversity ($\uparrow$) | |
| --- | --- | --- | --- | --- | --- |
| | Avg. Max. Toxicity | Toxicity Prob. | PPL | Dist-2 | Dist-3 |
| GPT2-large | 0.527 | 0.520 | 25.45 | 0.85 | 0.85 |
| PPLM | 0.520 | 0.518 | 32.58 | **0.86** | 0.86 |
| Non-toxic Expert | 0.485 | 0.464 | 40.61 | **0.86** | 0.86 |
| DAPT | 0.428 | 0.360 | 31.21 | 0.84 | 0.84 |
| GeDi | 0.363 | 0.217 | 60.03 | 0.84 | 0.83 |
| DExperts | 0.314 | 0.128 | 32.41 | 0.84 | 0.84 |
| **DETOXIGEN (GPT2-large)** | **0.254** | **0.115** | **27.54** | **0.86** | **0.86** |
| **DETOXIGEN (Llama-2-7B)** | **0.236** | **0.103** | **26.55** | 0.85 | 0.84 |

## 3.5 METRICS

**Toxicity** Following Gehman et al. (2020), we use the Perspective API[2] to measure the toxicity of generations. This score is obtained from a CNN model (Lecun et al., 1998) trained on a non-public corpus of Wikipedia comments. We compute two metrics based on the toxicity scores following Liu et al. (2021a): (1) Average Maximum Toxicity: The average maximum toxicity over $k = 25$ generations; (2) Toxicity Probability: The empirical probability of a generation with toxicity $\geq 0.5$ for at least once over $k = 25$ generations.

**Quality** The Quality metric consists of both fluency and diversity. Heeding both aspects makes it easier to spot cases where the generation is likely but generic, or diverse but unlikely. We use corpus-level Perplexity to evaluate fluency and Distinct-2 and -3 (Li et al., 2016) to evaluate diversity. Distinct-2 and distinct-3 correspond respectively to the number of distinct bigrams and trigrams divided by the total number of generated words.

## 3.6 BASELINE MODELS

We compare DETOXIGEN with a diverse set of previously reported baseline models (Gehman et al., 2020; Liu et al., 2021a), including Domain-Adaptive Pretraining (DAPT) (Gururangan et al., 2020), Plug-and-Play Language Models (PPLM) (Dathathri et al., 2019), Non-Toxic Expert (Liu et al., 2021a), Generative Discriminators (GeDi) (Krause et al., 2021), and Decoding-time Experts (DExperts) (Liu et al., 2021a). We follow these baselines to use Nucleus Sampling with $p = 0.9$ for generation.

# 4 RESULTS AND ANALYSIS

## 4.1 HYPERPARAMETER TUNING THROUGH VALIDATION SET

As mentioned in Section 3.3, we perform a grid search of $\alpha$ with values $1.0, 2.0, ..., 9.0$. We show the results on both GPT2-large and Llama-2-7b so that we can observe the trend on both early and more recent models. From Table 2, we can see that for both models, there is a steady increase in Perplexity (last column) as $\alpha$ grows, indicating a monotonic decrease in generation quality. Intuitively, this trend makes sense because the more we perturb the original output distribution, the more likely it is for the language model to generate less plausible tokens. To maintain a balance between toxicity and quality, we seek the tipping point where further increasing $\alpha$ only brings a diminishing return on reducing toxicity. We observe that for both models, this tipping point happens at $\alpha = 5.0$. Hence we adopt this hyperparameter setting throughout all other experiments.

---

[2]https://perspectiveapi.com/

Table 2: Validation results obtained by varying the *detoxifier* strength $\alpha$ from 1.0 to 9.0 with GPT2-large and Llama-2-7b. Each setting is evaluated on a held-out validation set of size 1k from REAL-TOXICITYPROMPTS. The boldfaced rows indicate tipping points where further increasing $\alpha$ starts to bring diminishing (sometimes even negative) returns on the balance between toxicity and fluency.

| Model | Alpha | Toxicity ($\downarrow$) | | Fluency ($\downarrow$) |
|---|---|---|---|---|
| | | Avg. Max. Toxicity | Toxicity Prob. | PPL |
| GPT2-large | 1.0 | 0.311 | 0.172 | 22.47 |
| | 2.0 | 0.284 | 0.145 | 23.54 |
| | 3.0 | 0.276 | 0.146 | 24.66 |
| | 4.0 | 0.261 | 0.127 | 25.83 |
| | **5.0** | **0.258** | **0.115** | **26.65** |
| | 6.0 | 0.261 | 0.128 | 27.54 |
| | 7.0 | 0.256 | 0.121 | 28.19 |
| | 8.0 | 0.257 | 0.125 | 28.82 |
| | 9.0 | 0.258 | 0.108 | 29.59 |
| Llama-2-7b | 1.0 | 0.290 | 0.160 | 19.88 |
| | 2.0 | 0.265 | 0.127 | 20.61 |
| | 3.0 | 0.252 | 0.108 | 21.20 |
| | 4.0 | 0.251 | 0.117 | 21.74 |
| | **5.0** | **0.241** | **0.104** | **22.31** |
| | 6.0 | 0.243 | 0.101 | 22.79 |
| | 7.0 | 0.241 | 0.106 | 23.13 |
| | 8.0 | 0.236 | 0.094 | 23.51 |
| | 9.0 | 0.233 | 0.097 | 23.88 |

## 4.2 RESULTS ON GPT2-LARGE

We then compare with previous approaches on GPT2-large. From Table 1, we can see that our model DETOXIGEN outperforms previous frameworks by a large margin although only tuned on the toxic split of the training data. Among all models, DETOXIGEN (GPT2-large) achieves the lowest Average Maximum Toxicity and Toxicity Probability, while obtaining a Perplexity that is quite close to that of the vanilla GPT-2 large, indicating minimum compromise on generation quality. The Llama-2-7B version of DETOXIGEN achieves even better results. However, it is based on a much stronger backbone language model, hence not comparable to previous work. We still include Llama-2 results in this table to show the gap between earlier and more recent large language models. We also follow Liu et al. (2021a) and report Distinct-N metrics, which are intended to prevent the model from degenerating into dull and generic continuations. We observe that the Distinct-N results do not vary much across diverse kinds of models. Hence for the results reported afterwards, we skip this metric and only report Perplexity.

## 4.3 ABLATION STUDIES ON MODEL SIZES

We also explore pairing models of different sizes as the *generator* and the *detoxifier*, respectively. This setting targets the cases where either latency is the major concern such that we want one small *detoxifier* to steer the generation of all other model sizes, or when we intend to train a *detoxifier* once and plug-and-play it with all other model sizes. The results of such pairings are presented in the matrix-like tables (Table 3, 5, 4, and 6). We report toxicity and quality in separate tables to make the comparisons clearer. From the four tables, we can observe quite a few interesting patterns.

**Consistent Toxicity Reduction** In the tables, we can observe that when comparing with the no-*detoxifier* setting (the column with *None* as header), our approach consistently and significantly reduces the toxicity of the backbone model while not sacrificing much on generation quality. This trend is observed for both the GPT-2 and the Llama-2 model families.

**Entries along the Diagonal** As shown in Table 3 and 4, entries on the diagonal of the result matrix (i.e., without the first column that has *None* as the header) consistently outperform their neighbors in terms of toxicity. These are the settings where the *generator* and the *detoxifier* share exactly the same backbone language model. They also achieve the best row-wise Perplexity as compared

Table 3: Toxicity results by pairing models of different sizes from the GPT-2 model family. All results are obtained on the validation set of size 1K. The column with the header *None* indicates that no *detoxifier* is used.

|  |  | *None* | *detoxifier* [Avg. Max. Toxicity \| Toxicity Prob.] | | | |
|---|---|---|---|---|---|---|
|  |  |  | GPT2-small | GPT2-medium | GPT2-large | GPT2-XL |
| *generator* | GPT2-small | 0.511 \| 0.413 | 0.264 \| 0.119 | 0.306 \| 0.161 | 0.318 \| 0.183 | 0.330 \| 0.195 |
|  | GPT2-medium | 0.514 \| 0.413 | 0.338 \| 0.195 | 0.280 \| 0.149 | 0.313 \| 0.182 | 0.331 \| 0.201 |
|  | GPT2-large | 0.499 \| 0.400 | 0.340 \| 0.215 | 0.322 \| 0.197 | 0.254 \| 0.115 | 0.314 \| 0.175 |
|  | GPT2-XL | 0.508 \| 0.432 | 0.352 \| 0.230 | 0.339 \| 0.202 | 0.313 \| 0.177 | 0.278 \| 0.124 |

Table 4: Toxicity results by pairing models of different sizes from the Llama-2 model family. All results are obtained on the validation set of size 1K. The column with the header *None* indicates that no *detoxifier* is used.

|  |  | *None* | *detoxifier* [Avg. Max. Toxicity \| Toxicity Prob.] | | |
|---|---|---|---|---|---|
|  |  |  | LLama-2-7B | LLama-2-13B | LLama-2-70B |
| *generator* | LLama-2-7B | 0.370 \| 0.285 | 0.241 \| 0.104 | 0.268 \| 0.131 | 0.287 \| 0.155 |
|  | LLama-2-13B | 0.371 \| 0.275 | 0.285 \| 0.143 | 0.248 \| 0.112 | 0.295 \| 0.164 |
|  | LLama-2-70B | 0.371 \| 0.276 | 0.295 \| 0.157 | 0.293 \| 0.167 | 0.277 \| 0.157 |

to off-diagonal models (Table 5 and 6). We hypothesize that this is because the output probability distributions of the *generator* and the *detoxifier* with the same underlying backbone parameters are more compatible with each other than backbones of different sizes. Recall in Section 1 that one of our major goals is to introduce as few new model parameters as possible. Our cross-model results clearly show that sharing weights between the *generator* and the *detoxifier* turns out to be the best setting among all we have investigated.

**Entries symmetric to the diagonal** Comparing entries that are symmetric to the diagonal (e.g., comparing GPT2-XL detoxified by GPT2-small with GPT2-small detoxified by GPT2-XL) in Table 3 and 4, we can observe a consistent pattern that given two models of different sizes, it is usually better to have the smaller model as the *generator* and the larger model as the *detoxifier* for detoxification. This indicates that larger models are more capable of capturing the distribution in the toxicity training corpus.

**Effect of Model Size Difference** From the toxicity tables, we can also observe that the larger the model size difference, the less effective the detoxification. For example, GPT2-XL detoxified by GPT2-small in Table 3 results in the worst toxicity among all settings, while we observe the same pattern where Llama-2-70B detoxified by Llama-2-7B has the highest toxicity among all settings.

## 5 DISCUSSION

It would be ideal if a *detoxifier* could work out of the box (plug-and-play) and be readily applied to any LLM *generator*, even with a different tokenizer. To achieve this, one can take a common subset of the vocabulary sets between the *generator* and the *detoxifier*, and only manipulate logits on this subset. We leave this as future work since the model families we investigate both already have diverse sizes.

Throughout the paper, we have been focusing on avoiding undesired attributes. However, we note that our framework can also be used to generate text with any desired style. All we need to do is flip the sign of the probability distribution correction term $\Delta P$ in Equation 2 and 3 as follows:

$$P(x_t|x_{<t}) = P_{GEN} + \alpha \Delta P \tag{7}$$
$$\Delta P = P_{CON} - P_{GEN}. \tag{8}$$

In addition, our approach could be applied to more general positive and negative attributes, including but not limited to politeness (Danescu-Niculescu-Mizil et al., 2013; Niu & Bansal, 2018), hate

Table 5: Quality results by pairing models of different sizes from the GPT-2 model family. All results are obtained on the validation set of size 1K. The column with the header *None* indicates that no *detoxifier* is used.

| | | *None* | *detoxifier* [PPL] | | | |
|---|---|---|---|---|---|---|
| | | | GPT2-small | GPT2-medium | GPT2-large | GPT2-XL |
| | GPT2-small | 49.90 | 60.46 | 76.83 | 82.90 | 91.02 |
| *generator* | GPT2-medium | 36.91 | 38.38 | 39.73 | 51.00 | 58.64 |
| | GPT2-large | 25.05 | 25.77 | 27.57 | 27.54 | 37.08 |
| | GPT2-XL | 18.16 | 18.54 | 18.76 | 19.77 | 19.80 |

Table 6: Quality results by pairing models of different sizes from the Llama-2 model family. All results are obtained on the validation set of size 1K. The column with the header *None* indicates that no *detoxifier* is used.

| | | *None* | *detoxifier* [PPL] | | |
|---|---|---|---|---|---|
| | | | LLama-2-7B | LLama-2-13B | LLama-2-70B |
| | LLama-2-7B | 19.65 | 22.94 | 23.12 | 22.35 |
| *generator* | LLama-2-13B | 21.69 | 28.38 | 25.47 | 26.90 |
| | LLama-2-70B | 22.39 | 29.68 | 28.68 | 26.55 |

speech (Golbeck et al., 2017), and microagressions (Breitfeller et al., 2019). In the case that we want to simultaneously control for multiple attributes, our framework is also compatible with mixed-batch inference (Liu et al., 2022a), where soft prompts of different attributes can be conditioned on in a single batch without increasing latency.

## 6 RELATED WORK

### 6.1 PARAMETER-EFFICIENT LEARNING

Parameter-efficient learning is a natural language processing paradigm to adapt a large language model to particular tasks or domains. It is usually used when fine-tuning the entire language model is prohibitively expensive. Among such approaches, LoRa (Hu et al., 2022) and AdaLoRa (Zhang et al., 2023) inject trainable rank decomposition matrices into each layer of the Transformer architecture, with the latter adaptively allocating the parameter budget among weight matrices according to their importance scores. Prefix Tuning (Li & Liang, 2021), P-Tuning (Liu et al., 2021b), and Prompt Tuning (Lester et al., 2021) prepend to the input sequence virtual tokens with trainable embeddings. Lastly, (IA)[3] scales activations by learned vectors. We choose Prompt Tuning in this work because it achieves competitive performance while involving no change in model architecture and not requiring any bootstrapping for the newly introduced model parameters.

### 6.2 CONTROLLABLE TEXT GENERATION

There have been multiple effective frameworks proposed for controllable text generation (Keskar et al., 2019; Sudhakar et al., 2019; Kurita et al., 2019; Welleck et al., 2020). [3] Among them, Domain-Adaptive Pretraining (DAPT) (Gururangan et al., 2020) and Self-Generation Enabled domain-Adaptive Training (SGEAT) (Wang et al., 2022) continues to finetune or apply parameter-efficient tuning to the backbone language model with a non-toxic subset of OpenWebText to adapt it to the non-toxic style. Plug-and-Play Language Models (PPLM) (Dathathri et al., 2019) trains a toxicity classifier and leverages gradients from that classifier to update the language model's hidden representations for each generation step. Generative Discriminators (GeDi) (Krause et al., 2021) prepends to the input a soft token serving as the label for the intended class of attribute. It can be viewed as prompt tuning with only one virtual token for each class (toxic and nontoxic). Decoding-time Experts (DExperts) (Liu et al., 2021a) train an expert and an anti-expert LM of opposing attributes

---

[3]We note that Inference-Time Policy Adapters (Lu et al., 2023) employs reinforcement learning for LM detoxification, but their approach assumes access to the Perspective API toxicity scores as a reward signal during training and hence not comparable to our work.

with full finetuning. During inference, the logits difference between the two experts serves as the correction term for the logits of the base language model. Our work is different from the previous approaches in that we adopt parameter-efficient tuning that only introduces a few trainable parameters and our training only requires toxic examples rather than examples from both classes.

## 6.3 Contrastive Decoding

Contrastive Decoding (Li et al., 2023; O'Brien & Lewis, 2023) is a search-based decoding method that optimizes a contrastive objective that returns the difference between the likelihood under a large and a small LM. Our algorithm is different from theirs in that we pair LMs of the same size (with the *detoxifier* further tuned) and perform all manipulations in the probability space rather than directly in the logits space.

## 7 Conclusion

We propose DETOXIGEN, a high-performing, parameter-efficient framework for detoxification during inference time. Our method only introduces a small portion of new parameters to train a *detoxifier* model that manipulates the output probability distribution of a generative model. On a standard detoxification benchmark, our approach outperforms all existing models in terms of both toxicity and quality. As language models grow ever larger in size, our controllable generation method shows the promise of quickly adapting to any language model without requiring additional computational resources or significant data curation efforts.

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
