# OpenReview forum: "Parameter-Efficient Detoxification with Contrastive Decoding"
_ICLR.cc/2024/Conference — Submitted to ICLR 2024_

### Official Review · Reviewer_qHFk · 2023-11-05

**Soundness:** 2 fair
**Presentation:** 3 good
**Contribution:** 2 fair
**Rating:** 5
**Confidence:** 4

**Summary:**

The authors propose a parameter efficient decoding time detoxification algorithm. The use a detoxifier, which is another generator that is finetuned on toxic data, to detect the toxic tokens and discount those generations by modifying the probability distribution of the generator. They show good detoxification results on the RealToxicityPrompts benchmark.

**Strengths:**

- The paper is well written and it is easy to follow.
- The detoxification results on RealToxocityPrompts is very good.
- Ablation studies on the model size is interesting.

**Weaknesses:**

- The authors claim that they  "are the first to apply parameter-efficient learning to controllable text generation for detoxification". However, there has been other work (such as ""Exploring the Limits of Domain-Adaptive Training for Detoxifying Large-Scale Language Models" by Wang et al) that use PEFT.
- One of the disadvantages of the proposed method is the cost as it needs both the generator as well as the detoxifier to do inference. The authors do not address this fact when they compare to other methods.
- The effect on fluency, measured by perplexity has been evaluated only on the RealToxicityPrompt dataset. A more diverse set will show boarder impact.
- The effect of the proposed approach on downstream tasks is not studied and it is not clear how the performance is affected.

**Questions:**

- In Table 1, what are the perplexity for the original model (alpha = 0)?

---

> ### Author Response · Authors · 2023-11-22
> **Response to Reviewer qHFk**
>
> We thank the reviewer for considering our paper well-written and the model performance very good.  We also appreciate the reviewer for noting that our ablation study on model size is interesting.
>
>
> * By using both the generator and the detoxifier during inference, the cost is higher as compared to other methods
>
> We respectfully disagree. In fact, cost is our approach’s strength rather than a weakness. Here we will compare our GPU memory usage and inference latency with the two top-performing baselines, namely GeDi or DExperts:
> (1) Our GPU memory usage is only around a third of the baselines because our detoxifier is trained with a parameter-efficient method and shares exactly the same backbone parameters as the generator (i.e., the LLM). This is different from the baselines which need to train an expert and an anti-expert LM of opposing attributes with full finetuning.
>
> (2) Our inference latency is only two-thirds of the baselines because our detoxifier only needs to rely on the toxic part of the training data. Hence instead of running the LLM three times for each prompt during inference (one for the LLM itself, one for the expert LM, and one for the anti-expert LM), our method only runs twice because there is no anti-expert involved.
>
> We sincerely hope that this detailed explanation alleviates the cost concern that may have negatively and greatly impacted our score.
>
>
> * In Table 1, what is the perplexity for the original model (alpha = 0)?
>
> That would be column 1 in Tables 6 and 7, where we present results with the vanilla generator (i.e., without the detoxifier).
>
>
> * Previous work has applied parameter-efficient learning to detoxification
>
> Thank you for pointing this out. Indeed Wang et al. (2022) applied parameter-efficient tuning to detoxification. We have toned down our claim in the revision. We do note a key difference though that their work assumes access to the Perspective API during training, while our work and the baselines we compare with do not, making the results not comparable to each other.
>
>
> * Evaluate the framework on more benchmark datasets and downstream tasks
>
> We selected the current evaluation benchmark because it makes it easier to compare with previous work that also focused on the same dataset. That said, we do agree that more benchmarks and downstream tasks can shed more light on the effectiveness of our approach.

---

### Official Review · Reviewer_EV4t · 2023-11-09

**Soundness:** 3 good
**Presentation:** 3 good
**Contribution:** 2 fair
**Rating:** 5
**Confidence:** 4

**Summary:**

This paper proposes a new way to detoxify language models using contrastive decoding, where the output probabilities of the base language model are negated by the probabilities of a language model trained on toxic data. The authors show that their techniques outperform a number of detoxification baselines for both toxicity reduction and fluency.

**Strengths:**

* The authors show that their technique enables toxicity reduction at many model sizes and for both GPT-2/LLaMA model families
* The technique is relatively straightforward and efficient

**Weaknesses:**

* The method seems like a pretty minor change from Liu et al 2021's DEXPERTS. As the authors note, their technique operate on the probabilities space, while the DEXPERTS technique operates in logits. Other than that, I can't find much difference. Their technique provides what looks like small gains over the DEXPERTS technique under their metrics. I would appreciate more analysis for why their formulation is preferable over DEXPERTS, and in which cases DEXPERTS might fail that their method would not.
* I would appreciate more qualitative examples of detoxification in the paper.
* I do not see mention of code release.
* Is perplexity the best measure of fluency? I would expect to see some human evaluations of generated text to confirm.
* I am not sure if most readers are familiar with the "Distinct-2/3" metrics of diversity, I would appreciate a brief explanation of this metric in the paper.

**Questions:**

Please see weaknesses above.

---

> ### Author Response · Authors · 2023-11-22
> **Response to Reviewer EV4t**
>
> We thank the reviewer for agreeing that our work leverages straightforward and efficient techniques and outperforms previous baselines.
>
>
> * DetoxiGen is the same as DExperts other than operating on the probability space
>
> We respectfully disagree. Operating on the probability space is only the difference between DetoxiGen and DExperts during inference. As summarized in the last paragraph of the Introduction section, there are two more major differences: (1) training efficiency: DetoxiGen adopts parameter-efficient training of the detoxifier while DExperts requires a full-finetuning of the language model; (2) data efficiency: DetoxiGen only requires training on the toxic data alone, where DExperts necessitates the use of non-toxic data as well.
>
> We sincerely hope that this detailed explanation alleviates the novelty concern that may have negatively and greatly impacted our score.
>
>
> * The approach overperforms previous work by a small margin
>
> We respectfully disagree. As Reviewer FrfU and qHFk both noted, our approach achieved superior performance over baselines. In terms of Avgerage Maximum Toxicity, we brought previous state-of-the-art from 0.314 all the way down to 0.254 with the same backbone model and data. This is an even bigger gap than that between GeDi and DExperts, not to mention that the lower the toxicity baseline, the harder it is to make any improvements.
>
>
> * No mention of a code release
>
> We thank the reviewer for bringing this up. We promised code release on Page 4 Footnote 1, and we have initiated the application process to open-source this project through our institution. We have migrated this footnote to page 1 in the revision to make it more salient.
>
>
> * Qualitative examples of detoxification
>
> We thank the reviewer for the suggestion. We did not include the qualitative examples due to ethical concerns as the outputs without detoxification could seem very offensive to the readers. In our code release, we plan to provide some sample toxicity-inducing prompts and include instructions for researchers and practitioners to compare the difference in outputs.
>
>
> * Effectiveness of perplexity as a measure of fluency
>
> We agree that perplexity alone may favor models that output generic, dull continuations that lead to low perplexity. That is the motivation that our work and previous baselines add the diversity metrics as a sanity check.
>
>
> * A detailed explanation of Distinct-N metrics
>
> We thank the reviewer for the suggestion. We did not elaborate on the Distinct-N metrics due to space constraints. We have added the details in the revision.

---

### Official Review · Reviewer_9Tow · 2023-11-10

**Soundness:** 4 excellent
**Presentation:** 4 excellent
**Contribution:** 3 good
**Rating:** 8
**Confidence:** 4

**Summary:**

The authors use a variant of contrastive decoding to generate non-toxic text. They do this by contrasting the outputs of the *generator* model with the *detoxifier* model which is soft-prompted to produce toxic text.

**Strengths:**

Originality: though this paper is not particularly original in its methods: it uses established NLP methods (contrastive decoding, soft-prompt tuning), it does apply them to non-toxic text generation which is fairly original.
Quality: The experiments and idea are straightforward and simple. I view this as a strength, since anything more elaborate would only muddy the waters.
Clarity: the paper itself is quite clearly presented, and I did not find any parts confusing.
Significance: Since the methods used are simple and general and the application useful, I think the proposed method has the potential to have significant impact.

**Weaknesses:**

While I respect the author's choice of sticking to a small set of reasonably chosen design decisions, I would have liked to trade some of the comprehensiveness on the model-size experiments for a broader look at some other hyperparameters, such as the method for creating the *detox* model (there are both more effective efficient fine-tuning methods like LoRA, and cheaper, more straightforward non-fine-tuning methods like plain-old prompting).

**Questions:**

No major questions.

---

> ### Author Response · Authors · 2023-11-22
> **Response to Reviewer 9Tow**
>
> We thank the reviewer for considering our work to have the potential to have a significant impact by introducing a straightforward and simple approach in a well-presented paper.
>
> * Explore other parameter-efficient tuning methods such as LoRa
>
> We went with Prompt Tuning as it is a more straightforward parameter-efficient approach that requires no change in model architecture (other than the few additional parameters from the soft token embeddings). That said, we agree that LoRa would be a nice baseline to compare with as part of the ablation studies. As the reviewer mentioned, due to space constraints, there is a trade-off on what studies to focus on — we figured that the experiments on model sizes would be very helpful for the audience to make design choices when applying our work.

---

> > ### Comment · Reviewer_9Tow · 2023-11-22
> >
> > Thank you for your reply. I don't see any need to change my evaluation.

---

### Official Review · Reviewer_FrfU · 2023-11-10

**Soundness:** 3 good
**Presentation:** 4 excellent
**Contribution:** 2 fair
**Rating:** 5
**Confidence:** 4

**Summary:**

This paper introduce a contrastive decoding methods with prompt tuning, which requires less parameters and performs better than a lot of works in the filed based on the result from realtoxicityprompts benchmark. The approach only requires toxic examples to train the detoxifier, without needing non-toxic contrastive data, making it more transferable. The framework could also steer generation towards desired attributes by flipping the probability manipulation.

**Strengths:**

the strengths:
- a lightweight framework that only requires toxic data for prompt tuning
- superior performance among six baselines.

**Weaknesses:**

- I am not sure how much I appreciate the technical contribution of this work, it seems to me that both of the findings from the generator and the detoxifier part are using an existing method, so it is hard to convince myself the novelty. However, it indeed proves how the framework works in the detoxification field, this is definitely valuable.
- the authors should show some qualitative examples to further back up table 2.
- Only one benchmark dataset is used.

**Questions:**

Please see the weakness parts.

---

> ### Author Response · Authors · 2023-11-22
> **Response to Reviewer FrfU**
>
> We thank the reviewer for noticing that our proposed method is lightweight and valuable as it leads to superior performance over previous baselines!
>
> * Justification of novelty
>
> As Reviewer 9Tow mentioned: although the components of our framework (contrastive decoding, parameter-efficient tuning) are already established methods, combining and applying them toward Detoxification is novel. As discussed in the last paragraph of the Introduction section, our approach makes the data curation much easier than previous work because it only requires toxic data, and parameter-efficient tuning also makes the amount of data required much smaller than full finetuning. This improved transferability (as described in Section 1, last paragraph) is also a significant part of the novelty. We sincerely hope that this explanation alleviates the novelty concern that may have adversely impacted our score.
>
> * Provide qualitative examples
>
> We thank the reviewer for the suggestion. We did not include the qualitative examples due to ethical concerns as the outputs without detoxification could seem very offensive to the readers. In our code release, we plan to provide some sample toxicity-inducing prompts and include instructions for researchers and practitioners to compare the difference in outputs.
>
> * Evaluate the framework on more benchmark datasets
>
> We selected the current evaluation benchmark because it makes it easier to compare with previous work that also focused on the same dataset. That said, we do agree that more benchmarks and downstream tasks can shed more light on the effectiveness of our approach.

---

### Author Response · Authors · 2023-11-22
**Response to all reviewers**

We thank the reviewers for their insightful feedback!

We also appreciate that overall, the reviewers consider our work lightweight, straightforward, valuable, well-performing, and clearly presented.

---

### Meta-Review · Area_Chair_9v2x · 2023-12-09

**Metareview:**

This paper proposes a contrastive-decoding method using a generator and another model prompt-tuned with a small set of toxic examples for detoxification. Results presented on realtoxicityprompts indicate better performance than baseline models.

**Strengths:** The method is efficient and works for models of different sizes. The paper is clearly presented and easy to follow for most readers.

**Weaknesses:** The central idea behind the paper is quite similar to other work on detoxification, even though the prompt-tuning idea makes the approach more efficient than Dexperts. Parameter-efficient learning for detoxification has also been previously explored, bringing into question what is unique about the findings of this paper. Moreover, for the specific approach proposed there has been not much exploration of hyperparameters or other parameter-efficient baselines. Finally, they consider only a single benchmark, going against the claim of “generality”.

Qualitative examples make any work on generation more comprehensive and give the readers an idea about the efficacy of the method beyond quantitative results which abstract away from more finer-grained findings. Due to the nature of this study, the authors claim they could not provide examples - but it’s unclear why toxic terms etc. were not redacted in the generations.

**Justification For Why Not Higher Score:**

See weaknesses above, somewhat thin novelty and empiricism, making the work not too exciting to the readers.

**Justification For Why Not Lower Score:**

N/A

---

### Decision · Program_Chairs · 2024-01-16

Reject